# Clinical and Epidemiological Characteristics of Neurocryptococcosis Associated with HIV in Northeastern Brazil

**DOI:** 10.3390/v15051206

**Published:** 2023-05-20

**Authors:** Ertênia Paiva Oliveira, Bruna Rodrigues de Sousa, Jucieli Firmino de Freitas, Rejane Pereira Neves, Moacir Batista Jucá, Paulo Sérgio Ramos de Araújo, Jailton Lobo da Costa Lima, Maria Amélia Vieira Maciel, Reginaldo Gonçalves de Lima-Neto

**Affiliations:** 1Postgraduate Program in Fungal Biology, Federal University Federal of Pernambuco (UFPE), Recife 50740-570, Pernambuco, Brazil; 2Hospital Correia Picanço, Department of Health from the State of Pernambuco, Recife 52060-060, Pernambuco, Brazil; 3Hospital das Clínicas, UFPE, Recife 50670-901, Pernambuco, Brazil; 4Department of Tropical Medicine, Center for Medical Sciences, UFPE, Recife 50670-901, Pernambuco, Brazil

**Keywords:** *Cryptococcus* sp, HIV risk factor, epidemiology, antiretroviral therapy

## Abstract

Cryptococcal meningitis is a serious infection of the central nervous system that is predominant in developing countries, caused by fungi of the genus *Cryptococcus*, and which affects immunosuppressed patients, especially those with HIV. Here, we aim to diagnose and characterize the clinical–epidemiological profile of cryptococcosis in patients admitted to two tertiary public hospitals in northeastern Brazil. The study is divided into three moments: (1) the isolation of fungus and diagnosis from biological samples collected between 2017 and 2019, (2) a description of the clinical and epidemiological characteristics of the patients, and (3) the experimental tests related to an in vitro susceptibility antifungal profile. The species were identified by MALDI-TOF/MS. Among the 100 patients evaluated, 24 (24.5%) were diagnosed with cryptococcosis based on positive culture. Clinical–epidemiological analysis showed a slightly higher prevalence in men between 30 and 39 years. When comparing the date of HIV diagnosis and the development of cryptococcosis, it was observed that 50% received the diagnosis of infection by cryptococcosis after or equal to a period of 12 months from being diagnosed with HIV; the other 50% received it within the first 30 days of the HIV diagnosis. Neurocryptococcosis was the most prevalent clinical form, and, at the time of hospital admission, the most common clinical signs were high fever (75%), intense headache (62.50%), and neck stiffness (33.33%). The cerebrospinal fluid showed 100% sensitivity and positivity for direct examination by India ink, and fungal culture. The mortality rate in this study was 46% (11/24), a lower rate than in the other literature. An antifungigram showed that 20 (83.33%) isolates were susceptible to amphotericin B and 15 (62.5%) to fluconazole. Mass spectrometry identified 100% of the isolates as *Cryptococcus neoformans*. In Brazil, this infection is not mandatory notifiable. Therefore, although there is little information on the subject, it is obsolete and does not express the reality of the facts, mainly in the northeast region, where this information is insufficient. The data obtained in this research contribute to the epidemiological knowledge of this mycosis in Brazil and will serve as a basis for future globally comparative epidemiological studies.

## 1. Introduction

Cryptococcosis is a systemic mycosis caused by a complex of pathogenic fungi included in the genus *Cryptococcus*, mainly *C. neoformans* and *C. gattii*. Among systemic mycoses, cryptococcosis is an opportunistic infection affecting mainly people living with human immunodeficiency virus (PLHIV). However, in uncommon cases, it can also affect healthy individuals, promoting latent and often asymptomatic infection [1]. According to the Brazilian Department of Epidemiological Surveillance of the Ministry of Health [2], cryptococcosis was reported as the most prevalent invasive fungal infection in terms of hospitalizations between 2000 and 2007, in addition to being associated with thousands of preventable deaths each year. It is noteworthy that there have been no data updates in Brazil since then.

Diagnosis of cryptococcosis is carried out by visualizing yeasts encapsulated in cerebrospinal fluid (CSF) by means of direct microscopic examination, and this reached 95% positivity using India ink [3]. At the same time, the culture must be performed to isolate the fungus, and this is considered the gold standard in laboratory diagnosis [4]. Nowadays, there is a wide variety of methods for DNA amplification using the polymerase chain reaction (PCR) test, all of which aim at genotyping in individual isolates, such as PCR-RFLP, PFGE, and RAPD [5]. Nowadays, ionizing proteins by a matrix-assisted laser and separating them by time-of-flight mass spectrometer (MALDI-TOF/MS) has been shown to be a fast and reliable tool for identifying fungi, including *Cryptococcus* species, in a clinical laboratory. This proteomic approach can replace phenotypic methods that demand more time for identification, in addition to providing an alternative to dependent and laborious genomic techniques [6,7].

The standard therapy used in the treatment of this systemic mycosis implies intravenous administration of amphotericin B (AmB) or its lipid derivatives and some azoles, especially fluconazole (FLZ), whether or not in combination with AmB [4]. However, this combination has generated resistance and recurrences, and although these are still sporadic, this has encouraged the use of in vitro susceptibility tests with a view of predicting the success of antifungal therapy and seeking other alternatives with antifungal potential [8].

Cryptococcal meningitis (CM) was first reported in 1890. However, in 1905 the first case of CM in humans was reported. With the beginning of antiretroviral therapy (ART), zidovudine was used in monotherapy between 1994 and 1996. However, new therapeutic regimens have been shown to decrease the viral load or even to make it undetectable, thus reducing morbidity and mortality. Although ART has been shown to be highly effective and safe, some patients may still progress to immunosuppression and become susceptible to opportunistic infections, such as cryptococcosis [9,10].

More research and knowledge about the disease are needed, mainly about epidemiological characteristics in Brazil, especially in the northeast. Therefore, this study set out to describe the clinical–epidemiological profile of cases of cryptococcosis in patients hospitalized in two tertiary public health units in the Brazilian Northeast and to identify and characterize clinical yeasts.

## 2. Materials and Methods

### 2.1. Design and Site of Study

This is a prospective and descriptive study with data collection from medical records and laboratory reports over 29 months. Moreover, it was also characterized by an experimental study with laboratory procedures performed from the set of strains. The study took place at the Correia Picanço Infectious Disease Hospital of the Department of Health of Pernambuco State (HCP/PE) and the Hospital das Clínicas at the Federal University of Pernambuco (HC/UFPE). The HCP/PE is a tertiary public health unit and reference for the treatment of infectious diseases with 170 beds. The HC/UFPE is a university teaching hospital with 418 beds, of which 26 are dedicated to infectious diseases.

### 2.2. Obtaining Clinical Yeasts from the Biological Samples

The collection of biological samples occurred from August 2017 to December 2019 by obtaining cerebrospinal fluid (CSF), blood, or urine from patients. Samples from the HC/UFPE were taken to the Laboratory for Research and Diagnosis in Tropical Diseases at the UFPE to be immediately processed. Direct microscopic examination and culture were performed as follows: CSF and urine were centrifuged (1500 to 2000× *g* for 5 min), then an aliquot of the sediment was placed onto a glass slide, and a drop of India ink was added, and subsequently, coverslips were placed on top of the slides. Microscopic analysis was carried out, and, at the same time, the biological material was cultured in Petri dishes with Sabouraud dextrose agar (SDA) plus 2 mL of chloramphenicol at 50 mg/L, which were incubated in duplicate at 35 °C in an aerobic atmosphere for up to 5 days. After growth, microscopic and macroscopic analysis of the cultures was carried out. Regarding the HCP/PE, previously isolated yeasts were obtained from the hospital’s laboratory [3,4,11].

### 2.3. Latex Agglutination Test

The *Cryptococcus* antigen latex test (Phadia Diagnósticos Ltd.a, Thermo Fisher Scientific, São Paulo, SP, Brazil) was used for two HCP/PE patients from whom urine was collected. This kit is a simple agglutination and screening test for the qualitative or semi-quantitative detection of the capsular polysaccharide antigen of *C. neoformans*. The Department of Health of the State of Pernambuco carries out this test only in its reference laboratory, called the Central Laboratory of the State, on urine samples from reference health units.

### 2.4. Clinical–Epidemiological and Demographic Features

The study included people with cryptococcosis living in the state of Pernambuco, defined as patients who presented a positive culture for *Cryptococcus* sp. Data were obtained from the Department of Archived Medical Records. The following variables were collected: (1) Demographic: gender, age, ethnicity, occupation, and spatial and temporal distribution of cases; (2) Clinical conditions: immunological status, length of hospitalization after the onset of signs and symptoms, the main cause of hospital admission, signs and symptoms referred, clinical manifestations, consciousness level, co-infections, and complications during hospitalization; (3) Laboratory tests: complete blood count, laboratory diagnosis (mycological examination), TCD_4_ lymphocyte count, and viral load count; (4) Treatment and evolution: antiretroviral and antifungal therapy, therapeutic approaches performed, and clinical evolution.

### 2.5. In Vitro Antifungal Susceptibility Testing

The reference microdilution assay, M27-A3, used was the Clinical and Laboratory Standards Institute (CLSI) one [12]. A standard Roswell Park Memorial Institute (RPMI) 1640 medium (Roswell Park Memorial Institute, Sigma Chemical Co., St. Louis, MO) without sodium bicarbonate was buffered to a pH of 7.0 ± 0.1 with 0.165 M of morpholino propane sulfonic acid (MOPS; Sigma-Aldrich, Cleveland, OH, USA). The RPMI 1640 medium was sterilized by filtration with a membrane 0.22 (Millipore, Darmstadt, Germany) and stored at 4 °C until use. Antifungal drugs amphotericin B (AmB) and fluconazole (FLZ) were diluted in dimethylsulfoxide (DMSO; Thermo Fisher Scientific, São Paulo, SP, Brazil) and distilled water, respectively. Ten different concentrations ranging from 0.03 to 16 μg/mL for AmB, and 0.125 to 64 μg/mL for FLZ, were used.

*Cryptococcus* isolates were subcultured on an SDA-added yeast extract and incubated at 35 °C for 48 h. Then, one to three single colonies were suspended in 5 mL of saline solution (0.85% sodium chloride), previously sterilized, and vortexed for 15 s. The cell suspension density was adjusted by a spectrophotometer at 530 nm and in 90% transmittance to obtain a final concentration of 5 × 10^5^ to 2.5 × 10^6^ CFU/mL. Each inoculum was two-fold-diluted to reach 10^3^ CFU/mL. *Candida parapsilosis* ATCC 22019 was used as the reference strain.

96-well flat-bottom microtiter plates (TPP, Trasadingen, Switzerland) were used. After the inoculum was added to the wells containing the serial dilutions of antifungal agents, the plates were incubated at 35 °C for 72 h to define the minimum inhibitory concentration (MIC). The MIC for FLZ was the lowest concentration, the reduction being ≥50% of the fungal growth considering the total growth from the control well, and for amphotericin B, the well that presented 100% inhibition was considered. The tests were performed in duplicate.

### 2.6. Proteomic Identification by MALDI-TOF MS

Spectra of clinical strains were obtained according to Lima-Neto et al. [13] with modifications. Protein was extracted from *Cryptococcus* isolates after 48 h of incubation on the SDA-added yeast extract. The MALDI TOF Autoflex III Mass Spectrometer (Bruker Daltonics Inc., Billerica, MA, USA), which was set up with a 1064 nm laser of neodymium crystal (Nd: Y_3_Al_5_O_12_) to 66% power, was used. The linear mode with a 104 ns pulsed laser and an acceleration voltage of +20 kV was used to register the mass range between 2000 and 20,000 Da. The peak lists obtained were exported to the software package MALDI Biotyper™ 3.1 (Bruker Daltonics, Bremem, Germany), which was used to achieve identifications.

## 3. Results

### 3.1. Laboratory Diagnosis

Ninety-eight patients with a diagnostic hypothesis of cryptococcosis were included in the study, seventy-six of them from the HC/UFPE and twenty-two from the HCP/PE. One hundred biological samples were collected, ranging from CSF to urine and blood. Among the 98 patients analyzed, 27 (27.5%) were diagnosed with cryptococcosis. Despite there being a total of 27 *Cryptococcus*-positive direct examinations and cultures, only 24 episodes of cryptococcosis were used to describe the clinical–epidemiological state. Three patients were excluded. Even though they presented yeasts encapsulated by the direct microscopic examination by India ink, they did not present a growth in culture, which is considered the gold standard.

Twenty-six biological samples from the 24 patients were evaluated: twenty (76.92%) CSF, four (15.38%) blood, and two (7.96%) urine (Table 1). Note that in two patients, two different biological samples, the CSF and urine, were collected for analysis. The latex agglutination test (crypto latex), used in both urine samples, had one negative result, even though the direct exam and culture had been positive from the CSF of the same patient.

### 3.2. Patients’ Characteristics

Clinical–epidemiological analyses were collected about the 24 patients diagnosed with neurocryptococcosis, cryptococcal meningitis, or systemic cryptococcosis who had DE and a positive culture. Fourteen (58%) of these patients were male, and ten (42%) were female.

Regarding age, the majority of patients (17/24; 70.83%) were between 30 and 49 years old (Table 2). Upon analyzing the males, there was evidence of an age group prevalence of those between 30 and 39 years old, and for the females, they were predominantly between 40 and 49 years old. Regarding ethnicity, six (25%) patients declared themselves to be black, four (16.7) were brown, and for the remaining fourteen (58.3%), the medical records did not contain ethnic specificities.

Among the 24 medical records analyzed, eleven patients (45.83%) had no reported occupation, three (12.5%) recorded household duties, two (8.32%) were farmers, two were car mechanics (8.32%), two (8.32%) were self-employed, and of the remaining four (4.16%), there was an academic assistant, a general services assistant, a bricklayer, and a journalist. Regarding the spatial distribution, 11 (45.84%) patients lived in the metropolitan region of Recife (the state capital of Pernambuco), and 13 (54.16%) lived in the countryside of the state.

When analyzing the temporal distribution of the cases, what became noticeable was that the highest incidence of hospitalization for cryptococcosis was in 2018, with thirteen (54.16%) cases, followed, in 2017, by six (25%) cases, and in 2019, there were three (12.5%) cases. Of these, nine (69.23%) in 2018 and four (66.66%) in 2017 were not using ART regularly.

When comparing the date of HIV diagnosis and the development of cryptococcosis, it was found that among the twenty patients diagnosed with HIV and cryptococcosis, one (5%) was diagnosed with HIV and cryptococcosis on the same day, four (20%) were diagnosed with cryptococcosis between 1 and 7 days after the HIV diagnosis, two (10%) between 15 and 30 days later, and one (5%) was found to have the infection between the 2nd and 12th month after the HIV diagnosis, while 10 (50%) received the diagnosis of infection by cryptococcosis after 12 months of their HIV diagnosis; one (5%) had already been living with HIV for 13 years before developing cryptococcosis, and in one (5%), there was no medical record that contained such information.

Regarding the time elapsed between the presence of the signs and symptoms related to cryptococcosis prior to their hospital admission, it was verified that 14 out of 24 (58.33%) of the patients presented symptoms seven or fewer days prior to hospitalization. In relation to hospital admissions, twenty-two out of twenty-four (91.66%) medical records showed that the patients were admitted to the hospital via the emergency service, of whom eight (36.36%) were transferred to the Intensive Care Unit (ICU).

The clinical signs and symptoms most recorded were high fever (18; 75%; 38 °C or higher), migraine (15; 62.5%), neck stiffness (8; 33.33%), diarrhea with or without blood (8; 33.33%), vomiting (7; 9.26%), weight loss (7; 29.26%), convulsion (6; 25%), cough (4; 16.66%), hearing impairment (4; 16.66%), visual impairment (3; 12.5%), nausea (3; 12.5%), fainting (3; 12.5%), loss of movement of the lower limbs (3; 12.5%), stomach distress (3; 12.5%), dyspine (3; 12.5%), hallucinations (2; 8.33%), sleepiness (2; 8.33%), diuresis (2; 8.33%), dizziness (1; 4.16%), and tremors (1; 4.16%). Regarding the level of consciousness, seventeen (70.83%) patients were conscious and oriented, five (20.83%) had a loss of level of consciousness, and in two (8.33%), there was no record of this symptom.

Cryptococcal meningoencephalitis was the most prevalent clinical form, and it was found in seventeen (70.83%) patients, followed by pulmonary cryptococcosis, also known as CID10–45.0, which was found in four (16.66%) patients. In one registry, one (4.16%) patient was reported to have neurocryptococcosis associated with the pulmonary form, and for two (8.33%) patients, there was no information recorded in the clinical record.

When evaluating the clinical symptoms presented by the two populations with cryptococcosis, the group with HIV reported the presence of bloody diarrhea and weight loss in comparison with the non-HIV group. No other clinical divergence between these two groups was observed.

The most prevalent co-infection was HIV/AIDS, totaling 20 (83.33%) cases. Among the co-infected HIV patients, 13 out of 20 (65%) presented other infections, such as tuberculosis (5; 25%), oral/esophageal candidiasis (4; 20%), syphilis (4; 20%), herpes simplex (3; 15%), hepatitis B (1; 5%), and neurotoxoplasmosis (1; 5%).

When evaluating the use of antiretroviral therapy (ART) by patients diagnosed with HIV, only four (20%) regularly used the medication, four (20%) had abandoned the treatment, one (5%) used it irregularly, six (30%) did not use it because they did not understand what HIV was, and in five (25%) cases, this piece of information was not included in their medical records. Nine had their TCD_4_ lymphocytes counted, including two having between 223 and 308 cells/mm^3^ and seven between 23 and 131 cells/mm^3^. The HIV viral load (CV) was counted in only eight patients, with values ranging from 1588 to 256,585 copies/mL.

Antifungal therapy was prescribed for 21 patients. One patient died within 24 h of hospitalization, and an antifungal drug had not been given. An antifungal induction phase by a combination regimen between AmB and FLZ was prescribed for 14 patients by means of intravenous administration of 50 mg/mL in 500 mL of a saline solution and 200 mg orally per day, respectively, for 6 weeks. FLZ alone was prescribed in six patients at a dosage of 400 mg/day for 12 weeks, and AmB alone in three patients intravenously in doses of 50 mg/mL in 500 mL of a saline solution for 8 weeks.

Some complications were observed during the hospitalizations, such as healthcare-related infection in nine (43%) patients, renal failure in four (19%), respiratory failure associated with mechanical ventilation in four (19%), blood transfusion due to severe anemia in three (14%), and deep vein thrombosis in one (5%) patient.

Regarding the patients’ clinical evolutions, it was verified that eleven (46%) patients were discharged with clinical improvement, of whom only four (36.36%) remained in the maintenance phase of cryptococcosis therapy, receiving fluconazole. Eleven (46%) patients died, and the clinical outcome was absent in the medical records of two (8%) patients.

### 3.3. Antifungal and Proteomic Assays

The CLSI-based antifungal susceptibility testing showed that twenty (83.33%) isolates were susceptible, and four (16.66%) were resistant to AmB. Regarding FLZ, it was observed that fifteen (62.5%) isolates were susceptible, eight (33.33%) were dose-dependent, and one (4.17%) was resistant (Table 3). According to the data collected from medical records, two of the four patients who showed AmB-resistant *Cryptococcus* isolates were treated with amphotericin B and died, and the other two received the therapeutic combination of AmB combined with fluconazole, and both were discharged from the hospital. As to the patient with FLZ-resistant *Cryptococcus* isolates, this patient received combined therapy with AmB and FLZ and was also discharged. Mass spectrometry using MALDI-TOF identified 100% (n = 24) of the isolates within species such as *Cryptococcus neoformans*. Fifteen (62.5%) were identified within subspecies such as *C. neoformans* var. *grubii*.

## 4. Discussion

Cryptococcal meningitis has a 70% mortality rate in low- and middle-income countries.^2^ The yeasts of the genus *Cryptococcus* are distributed worldwide, having as substrates air, soil, water, bird excreta, especially pigeons’, the surface and mucosa of animals, leaves, flowers, and decaying wood. In pigeon excretions, the fungus can remain viable for contagion for a period of up to two years [14,15].

This mycosis can affect both genders, but in men, its prevalence is around 70%, and in women, it is 30%, with still unclear reasons. The prevalence of age ranges varies. However, the highest is between 30 and 50 years old. These findings are fully in accordance with the literature, which states that cryptococcosis affects 5.1 times more men than women [1,16]. The number of cases in children has increased in recent years, which can be explained by the rise in the number of malnourished or immunocompromised children in this group. The statistics do not indicate differences in relation to ethnicity [14].

The few studies in Brazil show that the average age may suffer small variations, according to what was observed in two epidemiological studies at the Hospital of Uberlândia, Brazil, one conducted by Aguiar et al. [17], who reported an occurrence between 20 and 40 years old, and another in a public hospital in Rio Grande do Sul conducted by Mezzari et al. [18], where the most affected age group was the one between 30 and 39 years, corroborating our findings. Regarding ethnicity, a few studies report that there is a higher prevalence of mycosis in self-declared white patients, as described in the epidemiological study carried out in a teaching hospital in the state of Rio Grande do Sul by Ianiski et al. [18], and this is also reported by Mezzari et al. [17]. The lack of specifications in the medical records analyzed in our study, which showed that 58% (n = 14) of the cases did not present this piece of information, could explain this disagreement. The professional activities observed in this study were diverse, which made it impossible to correlate the work activity with cryptococcosis. Nevertheless, it was observed that among the jobs mentioned, only one was in a higher education level function. More studies should be thoroughly exploited to determine if they re-iterate our findings.

Regarding the spatial distribution, the municipality of Recife presents the highest incidence of cases of cryptococcosis, which may be explained by its being a metropolitan region that encompasses the largest healthcare hub in the Northeast of Brazil, as well as due to the patients living in the capital having greater access to hospitals in this region. In addition, Recife shelters have a high concentration of pigeons.

Our analysis on the patients’ immunological statuses with cryptococcosis is confirmed by other studies in the literature. Cryptococcosis is an opportunistic infection of medical importance, mainly in patients living with HIV/AIDS. In Brazil, cryptococcal meningitis is among the fourth most prevalent infections in PLHIV and the second most common in the central nervous system (CNS) [2,15]. According to the Brazilian AIDS Epidemiological Bulletin [2], the highest number of HIV cases is in the south and southeast regions (51.3% and 19.9%, respectively), and consequently, these also present the highest co-infections between HIV and cryptococcosis. In the northeast region, there has been a trend of growth in the last ten years, which needs to be supported by scientific evidence. The high correlation among cryptococcosis–HIV–tuberculosis co-infections in this study is corroborated by Mazzeri et al. [17], Fang et al. [19], and Oliveira et al. [2].

The lack of TCD_4_-counting was also described by Aguiar et al. [16]. The researchers evaluated 41 medical records, and only 17 (48.5%) presented the TCD_4_ count. Ianisky et al. [18] point out that low TCD_4_ (<100 cels/mm^3^) is likely the main predisposing factor to be infected with Cryptococcus and that neutocryptococcosis is closely connected with HIV patients.

Regarding signs and symptoms, the result obtained in this research is in accordance with the literature since cryptococcosis usually has nonspecific clinical conditions, making it hard to diagnose. In accordance with Cryptococcosis guidelines from the USA [4], the clinical form that predominates is the neurocryptococcosis (80%) isolate or is associated with lung involvement, as observed in our study that reported a case of this association. The second predominant clinical form is pulmonary. Firacative et al. [20], in Latin America, found that cryptococcal meningitis is still the most frequent clinical presentation, and the pulmonary form is the one that presents the most diagnostic delay due to its similarity in signs and symptoms with tuberculosis, thus resulting in high morbidity and mortality rates.

Studies carried out by Aguiar et al. [16] and Mezzari et al. [17] corroborate our findings since they observed an incidence of neurocryptococcosis of 75% and 74%, respectively, and that cryptococcal meningitis is still the most common clinical form. In the same studies, the pulmonary form is the one that presents the most diagnostic delay due to its similarity in signs and symptoms with tuberculosis, thus leading to a high rate of morbidity and mortality.

A study carried out in the state of the Brazilian Pantanal by Nunes et al. [14] performed an analysis on the use of ART and found that 41.6% (n = 35) of the population studied used ART, but only 20.2% of them used it regularly, and this percentage is to our findings. According to the literature, it is known that the regular use of ART by HIV patients will significantly decrease cryptococcal infection [19]. Carrijo et al. [10] argue that although treatment with ART is effective and reduces infections of opportunistic nature, it can still be verified that there is immunosuppression in some patients, even in those who use ART regularly. This was confirmed by our study. We found that four patients, even with regular use of ART, developed neurocryptococcosis. We attribute the many cases of death in this study to being related to the lack of use of ART, which was due to the lack of information about infection with HIV. Other deaths were due to non-adherence to treatment or due to irregular use of ART, culminating in the patients’ catching opportunistic infections.

In accordance with the guidelines of the Society of Infectious Diseases of America, in severe cases, the use of AmB associated with 5-fluocytosine (5FC) is recommended in the induction phase, followed by FLZ in the consolidation phase. However, 5FC is still unavailable in Brazil. Therefore, the association of AmB and FLZ during the induction phase is common in more severe cases. Hence, the therapeutic combination chosen, and its proportion in percentage terms, is in accordance with other studies found in the literature [4,19]. The mortality rate observed in our study was lower when compared to other recent studies, such as that by Ianiski et al. [19], who found 83% of lethality. Firacative et al. [20] studied cases in Latin America and stated that the lethality in this region is higher when compared to developed countries and that Brazil is ranked thirteenth in relation to death by cryptococcosis. Perhaps the smaller mortality rate presented in this research may reflect a rapid laboratory diagnosis associated with an effective therapeutic choice.

According to some studies, such as those by Soares et al. [21] and the Brazilian Cryptococcosis Consensus, *C. gattii* is an endemic species in Brazil, mainly in the north and northeast regions, and *C. neoformans* is more prevalent in the south and southeast regions. However, there was only isolation of the *C. neoformans* species in our study, which can be justified due to its being a species that affects immunocompromised patients the most, although in this same study, cases of non-HIV patients were related to *C. neoformans*. This implies that there is still much to be studied about the eco-epidemiology of this fungus.

## Figures and Tables

**Table 1 viruses-15-01206-t001:** Laboratory diagnosis for cryptococcosis.

Samples	CSF	Blood (Blood Culture)	Urine	Total	Positivity
	**n° %**	**n° %**	**n°%**		**100**
	20 76.92	4 15.38	27.7	26	
	(+) (−)	(+) (−)	(+) (−)		
Tests					
DE	20 0	4 0	0 0	24	100%
Culture	20 0	4 0	0 0	24	100%
Cryptolatex	0 0	0 0	1 1	2	50%

DE—direct examination; n°—Number; %—Percentage.

**Table 2 viruses-15-01206-t002:** Prevalence of cryptococcosis according to age group and gender of patients treated in health units in Pernambuco.

	Gender			
Age Group	Male		Female	
	n°	%	n°	%
**18–29**	3	21.4	1	10
**30–39**	7	50.0	2	20
**40–49**	3	21.4	5	50
**50–59**	1	7.2	1	10
**60–69**	0	0	1	10
**Total**	14	58	10	42

n°—Number; %—Percentage.

**Table 3 viruses-15-01206-t003:** Antifungal assay against clinical isolates of *Cryptococcus*.

	Drugs Evaluated	
Isolate	AmB ^††^	FLZ ^¥^
17	0.25	2
20	0.125	4
24	0.125	16
26	0.125	2
29	0.125	2
34	0.125	2
35	0.125	2
39	0.125	8
40	0.25	16
41	0.03	16
42	0.06	8
43	0.25	8
44	2	8
45	0.25	2
46	0.5	64
47	2	16
60	0.25	32
61	0.25	16
62	2	16
63	8	8
64	0.25	16
76	0.25	8
77	0.25	4
78	0.5	8
ATCC ^‡‡^	0.5	1

††AmB—Amphotericin B; ^¥^ FLZ—Fluconazole; ^‡‡^ ATCCs—American Type Culture Collection.

## Data Availability

Data sharing not applicable.

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
