# Peer review of "Clinical and Epidemiological Characteristics of Neurocryptococcosis Associated with HIV in Northeastern Brazil"

_viruses, 2023, doi:10.3390/v15051206_

Round 1
Reviewer 1 Report (Previous Reviewer 2)
Comments and Suggestions for Authors
No further comments
Reviewer 2 Report (Previous Reviewer 1)
Comments and Suggestions for Authors
Mr. Howard Wang,
Assistant Editor Viruses
Dear Mr. Wang,
The Manuscript ID: viruses-2116101 entitled “Clinical and epidemiological characteristics of neurocryptococcosis associated with HIV in northeastern Brazil” is an epidemiological description of several cases of cryptococcosis caused by the pathogenic yeast Cryptococcus neoformans in the State of Pernambuco, Northeast of Brazil. The manuscript was well improved since the last revision. I also reinforce for the editor the importance of publish Cryptococcus studies from Latin American region where rarely the antifungal drug flucytosine is available. All the questions raised by this reviewer have been answered. The manuscript is now suitable to be published in the journal Viruses.
Kind regards
Comments on the Quality of English Language
The English grammar is now suitable to published.
This manuscript is a resubmission of an earlier submission. The following is a list of the peer review reports and author responses from that submission.
Round 1
Reviewer 1 Report
Comments and Suggestions for Authors
Mr. Howard Wang,
Assistant Editor Viruses
Dear Mr. Wang,
The Manuscript ID: viruses-2116101 entitled “Clinical and epidemiological characteristics of neurocryptococcosis associated with HIV in northeastern Brazil” is an epidemiological description of several cases of cryptococcosis caused by the pathogenic yeast Cryptococcus neoformans in the State of Pernambuco, Northeast of Brazil. The manuscript has the potential to be published in the Journal Viruses due to the importance of this fungal infection in Latin American and the scarcity of data from this region of the country. Also, it is important to note that C. neoformans was recently included in the WHO fungal pathogens list (e.g. https://www.who.int/publications/i/item/9789240060241). In this reviewer opinion, the major pitfall of the manuscript the English grammar and the overall written style such as Portuguese letters within the text and references included in a non-standardized format. The authors are encourage to carefully revise the manuscript according to the Journal instructions, have the manuscript checked by a Native English Speaker and then resubmit it to the Journal.
Author Response
Dear Reviewer 1,
We completely agree with your observation. The manuscript has been revised according to the journal instructions, and checked by Native English Teacher with Degree in Language and Literature. All language adjustments are highlighted in blue in the text. In addition, the adjustments related to observations by reviewer 2 are highlighted in red.

Reviewer 2 Report
Comments and Suggestions for Authors
This is a very detailed investigation of clinical features of HIV related CNS cryptococcosis in northeastern Brazil. A total of 24 cases were included prospectively from August 2017 to December 2019. The results showed us the clinical-epidemiological characteristics of HIV-infected neurocryptococcosis in the Northeastern Brazil, contribute to the epidemiological knowledge of this disease in Brazil. But there are some issues as follows for further consideration.
1. Generally, as an epidemiological analysis, the small size of cases (24 cases) in this study limits drawing high quality points. So, I would like to focus on some specific topics not covering every aspect.
2. In the part of abstract line 30: I would like to add the exact mortality rate in this study.
3. In Materials and Methods line 102: It is unknow that only two patients were tested by Cryptococcus antigen latex test? Also, we commonly use for CSF and serum sample test. Here was only used for urine.
4. I would like to add the definition of CNS cryptococcosis in this part.
5. In the part of results line 153: Usually, at the first paragraph of results, we would like to describe the basic characteristics of all the enrolled patients. If there is no critically findings, it should be simplified.
6. Line 167: there is a little confused with this table (laboratory diagnosis for cryptococcosis). Totally a number of 24 cases of CNS cryptococcosis, 20 cases were all positive for CSF direct examination, that indicated that CSF direct examination was not performed in the other 4 cases. So, we want to know how can we make a diagnosis of CNS cryptococcosis for the 4 cases. Also, 4 patients with positive blood culture. So, would you let me know how many patients performed blood culture? Only 4 patients?
7. Line 192: the result told us that a patient from 2017 was non-HIV infected. But the title of this study showed us the neurocryptococcosis associated with HIV in northeastern Brazil. Please check it again?
8. Line 240: regarding the antifungal therapy, it is better to add the dosage and duration of AmB and FLZ.
9. Line 253: Four isolates were resistant to AmB, would be appreciate let us know that how about the antifungal use and outcome of these 4 patients. Also, 1 isolate was resistant to fluconazole, we would love to have the further information about the efficacy of FLZ.
10. In the part of discussion part Line 331: I am afraid that this paragraph may be better input ahead of signs and symptoms paragraph (line 321).
11. Line 348: This paragraph may also be at the ahead of signs and symptoms paragraph.
Author Response
Dear Reviewer 2
On behalf of all the authors, I would really like to thank the Reviewer #2 for their comments and suggestions regarding our manuscript. We are pleased to submit a revised version of the manuscript.
We have carefully considered the points raised by the reviewer and have modified our manuscript accordingly, creating a point-by-point letter addressing reviewer’s comments. All changes in the manuscript are highlighted in red. In addition, the English revision carried out Native English teacher with Degree in Language and Literature are highlighted in blue.
Yours sincerely,
Reginaldo Gonçalves de Lima-Neto
Professor of Medical Mycology and Tropical Medicine
Federal University of Pernambuco
Point-by-point response letter
- Generally, as an epidemiological analysis, the small size of cases (24 cases) in this study limits drawing high quality points. So, I would like to focus on some specific topics not covering every aspect.
- In the part of abstract line 30: I would like to add the exact mortality rate in this study.
R – This piece of information has been added.
- In Materials and Methods line 102: It is unknow that only two patients were tested by Cryptococcus antigen latex test? Also, we commonly use for CSF and serum sample test. Here was only used for urine.
R – The access to the kit for capsular antigen detection of C. neoformans is restricted to the Laboratory Central of the State of Pernambuco. Department of Health determined that only urine can be transported from Tertiary Health Units to this reference laboratory with medical justification. This piece of information has been added between lines 108-110.
- I would like to add the definition of CNS cryptococcosis in this part.
R – Definition for cryptococcosis has been added in line 113. In addition, the line 162 has been adjusted to reflect this definition.
- In the part of results line 153: Usually, at the first paragraph of results, we would like to describe the basic characteristics of all the enrolled patients. If there is no critically findings, it should be simplified.
R – From our point of view, describing the basic characteristics of patients will be repetitive with the text in subsequent topics. Also, most articles published in Viruses do not have this type of first paragraph in the Results section.
- Line 167: there is a little confused with this table (laboratory diagnosis for cryptococcosis). Totally a number of 24 cases of CNS cryptococcosis, 20 cases were all positive for CSF direct examination, that indicated that CSF direct examination was not performed in the other 4 cases. So, we want to know how can we make a diagnosis of CNS cryptococcosis for the 4 cases. Also, 4 patients with positive blood culture. So, would you let me know how many patients performed blood culture? Only 4 patients?
R - Allow me to clarify. Altogether, a number of 24 cases of cryptococcosis were diagnosed, 20 cases of CNS cryptococcosis by CSF direct examination and culture, and other 4 cases of fungemia by cryptococcosis through blood cultures. We cannot diagnose CNS cryptococcosis from these 4 cases. Even though, most of cases involve neurocryptococcosis, we extend our objective to cryptococcosis, as shown in lines 17 (abstract) and 78 (introduction) in order to cover all cases.
- Line 192: the result told us that a patient from 2017 was non-HIV infected. But the title of this study showed us the neurocryptococcosis associated with HIV in northeastern Brazil. Please check it again?
R – We make a mistake, and this sentence has been deleted.
- Line 240: regarding the antifungal therapy, it is better to add the dosage and duration of AmB and FLZ.
R – We agree. The regimens have been added in line 248-251
- Line 253: Four isolates were resistant to AmB, would be appreciate let us know that how about the antifungal use and outcome of these 4 patients. Also, 1 isolate was resistant to fluconazole, we would love to have the further information about the efficacy of FLZ.
R – We agree. The outcomes have been added in lines 265-270.
- In the part of discussion part Line 331: I am afraid that this paragraph may be better input ahead of signs and symptoms paragraph (line 321).
R – We agree and the paragraph has been repositioned.
- Line 348: This paragraph may also be at the ahead of signs and symptoms paragraph.
R – We agree and the paragraph has been repositioned.
